# HPV Type-Specific Prevalence a Decade after the Implementation of the Vaccination Program: Results from a Pilot Study

**DOI:** 10.3390/vaccines9040336

**Published:** 2021-04-01

**Authors:** Clara Fappani, Silvia Bianchi, Donatella Panatto, Fabio Petrelli, Daniela Colzani, Stefania Scuri, Maria Gori, Antonella Amendola, Iolanda Grappasonni, Elisabetta Tanzi, Daniela Amicizia

**Affiliations:** 1Department of Biomedical Sciences for Health, University of Milan, 20133 Milan, Italy; clara.fappani@studenti.unimi.it (C.F.); daniela.colzani@unimi.it (D.C.); mgorimaria@gmail.com (M.G.); antonella.amendola@unimi.it (A.A.); elisabetta.tanzi@unimi.it (E.T.); 2Department of Health Sciences, University of Milan, 20142 Milan, Italy; 3Department of Health Sciences, University of Genoa, 16100 Genoa, Italy; donatella.panatto@unige.it (D.P.); daniela.amicizia@unige.it (D.A.); 4School of Medicinal and Health Products Sciences, University of Camerino, 62032 Camerino, Italy; fabio.petrelli@unicam.it (F.P.); stefania.scuri@unicam.it (S.S.); iolanda.grappasonni@unicam.it (I.G.); 5Coordinated Research Center “EpiSoMI”, University of Milan, 20133 Milan, Italy

**Keywords:** HPV infection, urine samples, HPV genotype distribution, vaccine-preventable diseases, Italy

## Abstract

Human papillomavirus infection is a cause of the development of invasive cervical cancer. Three types of vaccine are currently available to prevent precancerous/cancerous lesions due to persistent infection, which is supported mainly by 7 different high-risk genotypes. The aim of this pilot study was to acquire preliminary data on type-specific prevalence 10 years after the implementation of the HPV vaccination program in Italy, in order to subsequently plan appropriate observational studies in the Italian population. First-voided urine samples were collected from 393 consenting subjects, both females and males, aged 18–40 years, and HPV DNA was detected by PCR amplification of a 450 bp L1 fragment. All amplified products were genotyped by means of the Restriction Fragment Length Polymorphism (RFLP) method. The female population was divided into three cohorts (“vaccine-eligible”, “pre-screening” and “screening” cohorts) according to the preventive intervention scheduled by age; males were included in the same three cohorts according to their year of birth. The overall prevalence of HPV infection was 19%, being higher in females than in males (22.1% vs. 13.6%, *p* = 0.03729). In the female population, 10 years after the start of the national immunization program, we observed a reduction in the prevalence of vaccine types and the number of circulating genotypes, especially in the “vaccine-eligible” cohort. The frequency of HPV vaccine types increased with age, particularly in males in the “pre-screening” and “screening” cohorts. Our study highlights the importance of monitoring HPV infection in both genders, to validate the effect of the HPV vaccination program.

## 1. Introduction

It is well known that human papillomavirus (HPV) infection is the most common sexually transmitted infection worldwide [1,2]. Soon after their first sexual experience, the majority of women acquire the HPV infection, which is generally transient and resolves without symptoms. However, persistent infection can cause a wide range of diseases, including benign lesions, precancerous lesions and cancers, in both women and men [3].

HPVs are small, double-stranded DNA viruses that infect the mucosa and epithelium. More than 200 HPVs have been fully characterized [4,5] and are classified into low- and high-risk types according to their carcinogenic potential [6]. HPV DNA is found in almost 100% of cervical cancer samples, and there is a consensus that persistent high-risk HPV infection is a necessary cause of cervical cancer [7,8]. HPV types 16 and 18 are clear, powerful carcinogens and cause approximately 70% of invasive cervical cancers. HPV16 is also detected in about 90% of HPV-related anal, vaginal, vulvar, penile and oropharyngeal cancers [9].

The low-risk types, HPV 6 and HPV 11, are responsible for about 90% of anogenital warts and benign/low-grade abnormalities in the genital district [10], and are also associated with recurrent respiratory papillomatosis, a proliferative disease of the upper aerodigestive tract [11].

Before the introduction of HPV vaccination, periodic Pap testing in women aged 25–65 years was the main way to prevent cervical cancer. Today, three prophylactic HPV vaccines targeting high-risk HPV types are available in many countries: 2-, 4- and 9-valent vaccines. All these prophylactic HPV vaccines protect against HPV-16 and HPV-18. Furthermore, the 4-valent HPV (4vHPV) vaccine also protects against HPV-6 and HPV-11, and the 9-valent HPV (9vHPV) formulation also against HPV-31, HPV-33, HPV-45, HPV-52 and HPV-58 [12].

Data from recent surveillance systems and studies demonstrate that HPV vaccines are effective in preventing infection and diseases (vaccine efficacy 93% to 100%) [13,14] and the real-life effectiveness of HPV vaccines has become increasingly evident, especially in females vaccinated before exposure to HPV. However, the vaccines are also effective in subjects who are already sexually active.

More than 70 countries, including 33 belonging to the WHO European Region, have introduced HPV vaccination into their national immunization programs for girls, and 11 countries have done so for boys [15].

In 2007, Italy initiated a national immunization strategy of active call and free-of-charge vaccination for girls aged 11–12 years [16,17]. Subsequently, in 2017, the strategy was extended to males aged 11–12 years (2006 cohort onwards) and at-risk subjects (men who engage in same-sex sexual behaviors) [17] following the latest evidence that HPV vaccination is also cost-effective in these target groups [16,17,18]. Furthermore, women aged 25 years who undergo cervical cancer screening are eligible for HPV vaccination, as recommended by the 2017–2019 National Vaccine Plan [17].

As yet, no comparison has been made between the prevalence of HPV before the introduction of the national HPV immunization program a decade ago and the current prevalence. The overall aim of the present study was to acquire preliminary data on type-specific prevalence 10 years after the implementation of the HPV vaccination program in Italy, in order to subsequently plan appropriate observational studies in the Italian population.

## 2. Materials and Methods

### 2.1. Ethics Statement

The study protocol, which was used by all research units, was approved by the Ethics Committee of the Liguria Region (LHU) Genoa, Italy (protocol n. 44/12). Written informed consent to participate in the study was obtained from all subjects.

### 2.2. Study Population

In accordance with the study protocol, we enrolled all consecutive eligible women and men aged 18–40 who spontaneously accessed sports medicine centers and universities located in three cities in different Italian Regions: Milan (Lombardy), Genoa (Liguria) and Camerino (Marche).

Females were divided into three cohorts according to their status regarding the prevention of HPV-related diseases: a vaccine-eligible cohort, which consisted of girls born between 1997 and 2001 and belonging to the first vaccination target cohort; a pre-screening cohort, comprising females born between 1991 and 1996, who were still too young to be called for a Pap test; and a screening cohort, consisting of girls born before 1990, who were eligible for a Pap test screening call.

Males were included in the different cohorts according to their years of birth. Recruitment was conducted between February 2017 and October 2019 and each participant was invited to provide an anonymous self-collected urine sample.

In Italy, the median age of sexual debut is 16 years in both sexes [19,20]. All participants in this study had already engaged in sexual activity. Indeed, being sexually active was an inclusion criterion.

### 2.3. Sample Collection and Treatment

First-voided urine samples were collected in sterile containers, kept at room temperature (RT) and processed within 6–8 h at each participating center. Briefly, 15 mL of each sample was centrifuged at 3800× *g* for 20 min at RT to obtain the cellular component. This phase was transferred to a new 1.5 mL collection test tube and centrifuged at 16,000× *g* for 15 min at RT. The pellet obtained was resuspended in 1 mL of PBS and aliquots were stored at −20 °C [21] until shipment to the center in Milan (Lombardy).

### 2.4. DNA Extraction

HPV DNA was extracted from 200 μL of concentrated urine samples by means of the NucliSENS^®^ easyMAG™ automated (bioMérieux bv, Lyon, France) method in accordance with the standard protocol with off-board lysis. The concentration and purity of extracted DNA were evaluated by means of a spectrophotometer (Thermo Scientific NanoDrop 2000; Thermo Fisher Scientific, Inc., Wilmington, DE, USA). DNA quality was assessed by amplifying a 268 bp (base pair) segment of the ubiquitous β-globin gene.

### 2.5. HPV DNA Detection and Genotyping

HPV DNA was detected through PCR amplification of a 450 bp L1 fragment by means of the degenerate primer pair ELSI-f and ELSI-r [21]. The amplification reaction conditions, after 5 min denaturation at 94 °C, involved 40 cycles of amplification. Each cycle consisted of a denaturation step (94 °C for 30 s), an annealing step (55 °C for 30 s) and an elongation step (72 °C for 30 s). The last cycle was followed by a 7 min elongation step at 72 °C. Each PCR run included positive (DNA extracted from HPV-16 positive cells, CaSki) and negative (water) controls. All amplified products were genotyped by means of the Restriction Fragment Length Polymorphism (RFLP) method, which is able to identify all HPV types present in the high-risk clade (HR-clade) and low-risk clade (LR-clade) of the alpha genus according to the latest IARC classification system (HR-clade, group 1: HPV-16, 18, 31, 33, 35, 39, 45, 51, 52, 56, 58, 59; group 2A: HPV-68; group 2B: HPV-26, 30, 34, 53, 66, 67, 69, 70, 73, 82, 85; LR types: HPV-6, 11, 28, 32, 40, 42, 43, 44, 54, 55, 57, 61, 62, 71, 72, 74, 81, 83, 84, 86, 87, 89) [22].

### 2.6. Sample Size Calculation

The sample size was calculated according to data published by the HPV Information Centre [23], to estimate differences in the prevalence of HPV infection by age (<25 and 25–44 years of age). Taking into account the lack of information on HPV infection in the general male population and the sexual transmission route, we considered an HPV prevalence in males comparable to that in females.

Overall, an estimated prevalence of 15% in males and females was considered: 18% in subjects aged <25 years and 11% in those aged 25–40 years [23].

The sample size was calculated by means of the following formula, considering a 95% confidence interval (95% CI) with an absolute precision of ±5%:$$Sample=\frac{1.96^{2}\times HPVprevalence\;estimated\times \left(1-HPVprevalence\;estimated\right)}{absolute\;precision^{2}}$$

The sample size of the overall population was estimated to be 196 subjects, for the population under 25 years was 227, and for those between 25 and 40 years of age was 150 subjects enrolled.

### 2.7. Statistical Analysis

As the HPV vaccination strategy was the same across the country, with no differences among regions, a comprehensive analysis was conducted.

Descriptive statistics (median and IQR) were applied to describe the entire study sample. HPV prevalence was expressed as a crude proportion with corresponding 95% confidence intervals (95% CI) and calculated by means of the Mid-p exact test, assuming a normal distribution. A *p*-value < 0.05 was considered statistically significant (2-tailed test). Statistical analysis was performed by means of OpenEpi, version 3.01 [24].

## 3. Results

A total of 393 subjects participated in this study and provided an anonymous self-collected urine sample. The study population included 253 (64.4%) females and 140 (35.6%) males and the median age was 23 years (range: 18–40). Of the 393 participants, 220 were recruited in Milan, 115 in Genoa and 58 in Camerino (Table 1).

Of the 393 participants, 75 (19.08%, 95% CI: 15.43–23.2) tested HPV DNA-positive: 56 of the 253 (22.13%, 95% CI: 17.34–27.56) females and 19 of the 140 (13.57%, 95% CI: 8.62–20.01) males (*p* = 0.03729).

No difference in HPV prevalence was observed among the female subjects in the different cohorts, while an increase in positivity was observed in males: from 0% in the vaccine-eligible cohort to 12.28% in the pre-screening cohort and 19.67% in the screening cohort. HPV prevalence rates between the sexes and across cohorts are reported in Table 2.

Of the 75 HPV DNA-positive samples, 4 (5.33%) were not typable (NT). Overall, 111 different infections sustained by 34 different genotypes were identified. A single genotype was detected in 71.83% (51/71; 95% CI: 60.56–81.35) of the HPV DNA-positive samples, and 74.51% (38/51; 95% CI 61.28–85.03) of the genotypes identified belonged to the HR clade. The remaining 20 HPV DNA-positive urine samples (28.17%; 95% CI 18.65–39.44) showed multiple infections, and 95% (19/20; 95% CI: 77.72–99.75) of these were sustained by at least one genotype belonging to the HR clade. Multiple infections were detected in 16 of 56 (28.57%) infected females and in 4 of 19 (21.06%) infected males. HPV infections and HPV infecting types, broken down by the three different cohorts, are shown in Table 3. No difference among the cohorts was observed with regard to the prevalence of HPV infections, the frequency of multiple and single infections, or the prevalence of any type. The type-specific HPV prevalence in the 393 subjects, broken down by cohorts, is shown in Figure 1.

The numbers of HR and LR types observed were lower in the vaccine-eligible cohort than in the pre-screening and screening cohorts, though HPV-26, HPV-30, HPV-54 and HPV-72 were only detected in this cohort and a higher prevalence of HPV-52, HPV-56 and HPV-59 was observed.

The prevalence of vaccine types was lower in the vaccine-eligible cohort than in the other cohorts. Indeed, the frequency of HPV vaccine types was significantly lower in the vaccine-eligible cohort than in the screening cohort (*p* = 0.02931, Table 4). The frequency of HPV vaccine types, regardless of vaccine formulation, increased with age, particularly in the male population belonging to the pre-screening and screening cohorts.

## 4. Discussion

Human papillomavirus is the most common sexually transmitted infection worldwide. The Centers for Disease Control and Prevention (CDC) estimate that at least half of all sexually active individuals will acquire HPV at some point in their lives and that at least 80% of women will acquire an HPV infection by the age of 50 [25].

While most HPV infections do not cause symptoms and resolve spontaneously, persistent infection may lead to the development of a range of conditions of the reproductive tract, including precancerous lesions that may progress to cancer, both in women and men.

Human papillomavirus is a necessary (but not sufficient) cause of invasive cervical cancer. Three vaccines (2-valent, 4-valent and 9-valent) with prophylactic indications are currently available to prevent precancerous lesions and cancers due to persistent infection supported by up to 7 different high-risk genotypes (HPV-16, HPV-18, HPV-31, HPV-33, HPV-45, HPV-52, HPV-58). The 4-valent and 9-valent vaccines also offer protection against genital warts and low-risk lesions caused by HPV-6 and HPV-11 [12].

Since 2008, HPV vaccination has been offered free of charge to 12-year-old girls in Italy, and 70.6% of eligible subjects belonging to the first vaccinated cohort (females born in 1997) have received at least one vaccine dose [26]. The high coverage rate of HPV vaccination has been shown to impact on HPV-related diseases, reducing the burden of HPV infection, genital warts and cervical disease [27,28,29,30].

In the present study, the prevalence of HPV infection was evaluated in urine samples collected in 2017–2019 from subjects of both sexes. Cervical brushing still is the gold standard for HPV detection; indeed, a partial discordance between paired urine and cervical samples was observed in PCR-based genotyping results. Nevertheless, urine sampling is a highly sensitive method and a viable alternative to cervical brushing in women and urethral swabs in men, in order to detect and characterize HPV infection. Moreover, urine sampling is easy to carry out and offers the unquestionable advantage of non-invasiveness and the possibility of reaching a high adherence rate among women, men and vulnerable populations in the youngest age groups. Therefore, several studies have suggested that urine-based HPV detection is a suitable and effective tool for both epidemiological surveillance and screening [21,31].

The worldwide prevalence of HPV among women with no evidence of cervical lesions has been estimated at around 12% [32]. Our results showed an HPV prevalence of 22.1% in women aged 18–40 years, which is higher than that reported in the pre-vaccination era in the same area (14.3%) [33,34], with no significant differences among cohorts (23.1% in vaccine-eligible, 22.9% in pre-screening, and 20.3% in screening cohorts).

To date, few data are available on the epidemiological status of HPV infection in males without clinical manifestations. A review published prior to the introduction of HPV vaccination in men found an HPV prevalence among sexually active men ranging from 1% to 84% worldwide. In a study of men aged 19–82 years (median age 46 years) from Brazil, Columbia, Thailand, the Philippines and Spain, the HPV prevalence was 16% [35]. In the European low-risk population, the highest prevalence (34%) was observed in Denmark, while in Italy the overall prevalence was 9% [35]. Our data, collected from 140 men aged 18–40 years, showed an overall HPV prevalence of 13.6%, with higher values in those male cohorts other than the vaccine-eligible cohort; indeed, we observed that HPV prevalence increased with age (from 0% in the vaccine-eligible cohort to 12.3% in the pre-screening cohort and 19.7% in the screening cohort).

Our data showed a higher prevalence of HPV infection in females in the vaccine-eligible cohort (23.1%) and in males in the screening cohort (19.7%). These findings are in agreement with those of a systematic review of the prevalence of genital HPV in men over 18 years of age, in which this prevalence was seen to peak in older men, but not in older women [35].

Ten years after the introduction of HPV vaccination for the female population, HPV-16 is still the most frequently observed genotype [36]. However, in our vaccine-eligible cohort, the most frequent genotype was HPV-52, which, in the pre-vaccine era was the second most frequent genotype after HPV-16 [36,37,38]. Notably, the frequency of HPV-52 was twice that of HPV-16. As HPV-52 is one of the types contained in the 9vHPV vaccine formulation, a reduction could be expected in the coming years.

It is noteworthy that a reduction in the number of HR-HPV and in the circulation of HPV vaccine types (2v, 4vHPV) was observed in the vaccine-eligible cohort in comparison with previous studies conducted in the same area in the pre-vaccine era [36,39]. No vaccine types (2v, 4v, 9vHPV) were found in males belonging to this cohort. Furthermore, it is important to point out that, as the vaccine-eligible cohort consisted of women with unknown vaccination status and unvaccinated males, the circulation of vaccine-preventable genotypes was expected.

In the other cohorts, the frequency of HPV vaccine types (2v, 4v, 9vHPV) increased with age in both sexes, but the preventable fraction remained around 90%. These data suggest that catch-up programs should be implemented in order to vaccinate young women and young men beyond the 12-year-old cohort. The highest prevalence of vaccine types was observed in the screening cohort, especially in the male population, and this finding underlines the importance of including men in immunization programs.

## 5. Conclusions

In this study, we observed an overall prevalence of genital HPV infection of 19%, which was higher in females than in males. In the female population, ten years after the start of the national immunization program, we observed a reduction in the prevalence of vaccine types and also in the number of circulating genotypes, especially in the vaccine-eligible cohort. The absence of vaccine types in the male vaccine-eligible cohort could be evidence of a herd effect, despite suboptimal vaccination coverage (about 70% with three doses).

None of the men in our study had been vaccinated (the male vaccination program started in 2017 for 12-year-olds); our data are therefore essential to understanding the future effects of vaccination in this population.

The limit of this study is that the number of subjects enrolled was not large enough to be fully representative of the Italian population, and hence to precisely assess the impact of vaccination in Italy. However, this was a pilot study to evaluate the suitability of urine sampling in a monitoring context and in a larger population.

Our study highlights the importance of vaccinating both women and men in order to achieve good vaccination coverage and to protect males from HPV-related diseases. It also underlines the importance of including catch-up vaccination for women up to age 26 in the national immunization program, an approach that could be implemented by taking advantage of the Pap test call.

## Figures and Tables

**Figure 1 vaccines-09-00336-f001:**
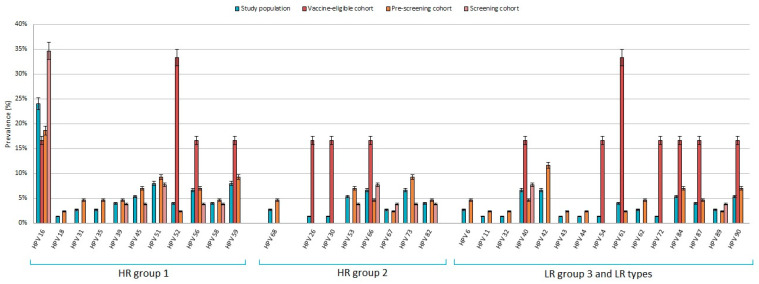
Type-specific prevalence with 95% confidence intervals in urine samples from 393 subjects, broken down by cohorts: vaccine-eligible, pre-screening and screening cohorts.

**Table 1 vaccines-09-00336-t001:** Baseline characteristics of patients enrolled from sports medicine centers and universities located in three cities in different Italian Regions: Milan (Lombardy), Genoa (Liguria), and Camerino (Marche) in Northern/Central Italy.

	Median Age (IQR)	Females *N* (%)	Males *N* (%)	Vaccine-Eligible Cohort *N* (%)	Pre-Screening Cohort *N* (%)	Screening Cohort *N* (%)
Camerino *N* = 58	23 (22–24)	25 (9.9%)	33 (23.6%)	3 (6.3%)	46 (21.4%)	9 (6.9%)
Genoa *N* = 115	21 (20–22.5)	115 (45.5%)	0	4 (8.3%)	99 (46.1%)	12 (9.2%)
Milan *N* = 220	27 (22–33)	113 (44.7%)	107 (76.4%)	41 (85.4%)	70 (32.6%)	109 (83.8%)
Total	23 (21–29)	253(64.4%)	140 (35.6%)	48 (12.2%)	215 (54.7%)	130 (33.1%)

**Table 2 vaccines-09-00336-t002:** Prevalence of positive subjects, broken down by gender and cohort: vaccine-eligible cohort (born between 1997 and 2001), pre-screening cohort (born between 1991 and 1996) and screening cohort (born before 1990). A significant *p*-values are indicated in bold (significance considered *p* < 0.05).

Cohort	Males		Females			Total	
	HPV-DNA+/N (%)	95% CI	HPV-DNA+/N (%)	95% CI	*p*	HPV-DNA+/N (%)	95% CI
Vaccine-eligible	0/22	0.0–12.73	6/26 (23.08)	9.92–41.95	**0.01876**	6/48 (12.50)	5.86–24.70
Pre-Screening	7/57 (12.28)	5.53–22.79	36/158 (22.78)	16.75–29.81	0.08748	43/215 (20.0)	15.06–24.19
Screening	12/61 (19.67)	11.12–31.05	14/69 (20.29)	12.04–30.99	0.9338	26/130 (20.0)	13.79–27.53
Total	19/140 (13.57)	8.62–20.01	56/253 (22.13)	17.34–27.56	**0.03729**	75/393 (19.08)	15.43–23.2

**Table 3 vaccines-09-00336-t003:** HPV infections and HPV infecting types, broken down by cohorts: vaccine-eligible cohort (born between 1997 and 2001), pre-screening cohort (born between 1991 and 1996) and screening cohort (born after 1990).

**HPV DNA-Positive Subjects**						
	**Vaccine-Eligible Cohort, *N* = 6**		**Pre-Screening Cohort, *N* = 43**		**Screening Cohort, *N* = 26**	
**HPV Infections**	**N (%)**	**95% CI**	**N (%)**	**95% CI**	**N (%)**	**95% CI**
Single infections	3 (50.0)	14.66–85.34	28 (65.11)	50.08–78.17	20 (76.92)	58.05–90.08
Multiple infections	3 (50.0)	14.66–85.34	15 (34.88)	21.83–49.92	2 (7.69)	1.31–23.16
NT	0	0–39.3	0	0–6.73	4 (15.38)	5.09–33.06
**Total Number of HPV Infections**						
	**Vaccine-Eligible Cohort, *N* = 16**		**Pre-Screening Cohort, *N* = 71**		**Screening Cohort, *N* = 24**	
**HPV Infecting Types**	**N (%)**	**95% CI**	**N (%)**	**95% CI**	**N (%)**	**95% CI**
HR Group 1	5 (31.25)	12.46–56.32	32 (43.66)	32.49–55.34	15 (62.5)	15.81–38.3
HR Group 2	3 (18.75)	5.00–43.01	13 (18.31)	10.58–28.58	6 (25)	5.10–21.82
LR Group 3 and other LR types	8 (50.0)	28–7226.59–73.41	26 (36.62)	26.05–48.26	3 (12.5)	3.28–30.36

**Table 4 vaccines-09-00336-t004:** Proportion of subjects free from high-risk HPV vaccine types (2-, 4- and 9-valent vaccines) broken down by cohort and gender.

**Study Population**						
**Proportion of Uninfected Subjects**	**2vHPV** **(HPV-16 and HPV-18)**		**4vHPV** **(HPV-16, HPV-18, HPV-6 and HPV-11)**		**9vHPV** **(HPV-16, HPV-18, HPV-31, HPV-33, HPV-45, HPV-52, HPV-58, HPV-6 and HPV-11)**	
**Cohort**	***N* (%)**	**95% CI**	***N* (%)**	**95% CI**	***N* (%)**	**95% CI**
Vaccine-eligible, *N* = 48	47 (97.92)	89.1–99.63	47 (97.92)	89.1–99.63	45 (93.75)	83.17–97.85
Pre-screening, *N* = 215	206 (95.81)	92.24–97.78	203 (94.41)	90.5–96.78	195 (90.70)	86.07–93.9
Screening, *N* = 130	121 (93.08)	87.37–96.32	121 (93.08)	87.37–96.32	119 (91.54)	85.48–95.21
**Female Population**						
**Proportion of Uninfected Females**	**2vHPV**		**4vHPV**		**9vHPV**	
**Cohort**	***N* (%)**	**95% CI**	***N* (%)**	**95% CI**	***N* (%)**	**95% CI**
Vaccine-eligible, *N* = 26	25 (96.15)	81.11–99.32	25 (96.15)	81.11–99.32	23 (88.46)	71.03–96
Pre-screening, *N* = 158	151 (95.57)	91.14–97.84	148 (93.67)	88.74–96.53	142 (89.87)	84.18–93.67
Screening, *N* = 69	66 (95.65)	87.98–98.51	66 (95.65)	87.98–98.51	66 (95.65)	87.98–98.51
**Male Population**						
**Proportion of Uninfected Males**	**2vHPV**		**4vHPV**		**9vHPV**	
**Cohort**	***N* (%)**	**95% CI**	***N* (%)**		***N* (%)**	**95% CI**
Vaccine-eligible, *N* = 22	22 (100)	85.14–100	22 (100)	85.14–100	22 (100)	85.14–100
Pre-screening, *N* = 57	55 (96.49)	88.08–99.03	55 (96.49)	88.08–99.03	53 (92.98)	83.3–97.24
Screening, *N* = 61	55(90.16)	80.16–95.41	55 (90.16)	80.16–95.41	53 (86.89)	76.2–93.2

## Data Availability

The data presented in this study are available on request from the corresponding author.

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
