# Peer review of "HPV Type-Specific Prevalence a Decade after the Implementation of the Vaccination Program: Results from a Pilot Study"

_vaccines, 2021, doi:10.3390/vaccines9040336_

Round 1
Reviewer 1 Report
In this work, the authors analyse the HPV type specific prevalence in 3 different cohorts of patients in order to evaluate the impact of HPV vaccination program in Italy. They collect 393 urine samples to be analysed by PCR for HPV DNA detection followed by RFLP methods for genotyping. A reduction in the prevalence of vaccine types and in the number circulating genotypes was observed, particularly in the cohort of vaccine eligible women.
There is no report evaluating the impact of HPV vaccination program in Italy except for a very recent study (Bosco et al., SciRep 2021), therefore the potential relevance of this study is very high; also, results shown in this work are interesting and very encouraging. However, there are a series of main weaknesses to be highlighted:
- Despite collected from different centers, the number of samples is too small for a comprehensive and effective analysis of the impact of a national vaccination program as intended by the authors themselves in the title. For instance, in Bosco et al., SciRep 2021, approximatively 1000 samples are analysed in the only region of Sicily. Therefore, a much higher number of samples should be analysed to be representative of the impact of vaccination in the whole Country.
- Authors correctly state that urine test provide some advantages in terms of collection of samples and, consistently, employ a PCR-RFLP method previously shown having high agreement between urine and vaginal samples (Tanzi et al, 2012). However, there are also several reports showing discordance between the two methods of analysis and the number of collected samples along with the type of test appear to be of high relevance for this evaluation. Thus, for a pilot study evaluating impact of vaccination in a Country, urine samples paired with swabs would be would have been much reliable.
On the base of these observations, this referee believes that the body of the work is too weak to deserve publication. On the other hand, this referee encourages the authors to improve this promising study in order to build a robust paper fully deserving publication.
Author Response
Reply to Reviewer 1
# REV1
We thank the Reviewer #1 for his comments. We agree with the Reviewer #1 and we have made almost all of the suggested corrections.
In this work, the authors analyse the HPV type specific prevalence in 3 different cohorts of patients in order to evaluate the impact of HPV vaccination program in Italy. They collect 393 urine samples to be analysed by PCR for HPV DNA detection followed by RFLP methods for genotyping. A reduction in the prevalence of vaccine types and in the number circulating genotypes was observed, particularly in the cohort of vaccine eligible women.
There is no report evaluating the impact of HPV vaccination program in Italy except for a very recent study (Bosco et al., SciRep 2021), therefore the potential relevance of this study is very high; also, results shown in this work are interesting and very encouraging. However, there are a series of main weaknesses to be highlighted:
Despite collected from different centers, the number of samples is too small for a comprehensive and effective analysis of the impact of a national vaccination program as intended by the authors themselves in the title. For instance, in Bosco et al., SciRep 2021, approximatively 1000 samples are analysed in the only region of Sicily. Therefore, a much higher number of samples should be analysed to be representative of the impact of vaccination in the whole Country.
We agree with Reviewer #1, the study population included only 393 subjects, but this should be viewed as a pilot study that will expand the number of subjects enrolled in the future by reaching a larger sample size. This limit was added in the Conclusion section of the manuscript.
However, we think our preliminary results are relevant and encouraging in terms of Public Health.
Authors correctly state that urine test provide some advantages in terms of collection of samples and, consistently, employ a PCR-RFLP method previously shown having high agreement between urine and vaginal samples (Tanzi et al, 2012). However, there are also several reports showing discordance between the two methods of analysis and the number of collected samples along with the type of test appear to be of high relevance for this evaluation. Thus, for a pilot study evaluating impact of vaccination in a Country, urine samples paired with swabs would be would have been much reliable.
We agree with the reviewer’s comment that several studies report a discordance between the urine and the cytological pap smear as a test sample. However, Tanzi et al. evaluated the sensitivity and specificity of the urine testing compared to the conventional cervical smear testing using a PCR-based method and found that this method proved highly sensitive and specificity for HPV-DNA detection and genotyping in urine samples. A urine-based test therefore represents a suitable and effective tool for epidemiological surveillance, screening programs in vulnerable populations and for monitoring HPV infection in young women who are unsuitable for conventional screening based on cervical sampling. Furthermore, the urine test appears to be useful for HPV detection in males, as a gold standard among sampling methods is not available for these subjects
Other published studies, performed by the same research group, have used urine sampling as a tool to detect and characterize HPV infection in different populations:
à Tanzi E, Bianchi S, Fasolo MM, Frati ER, Mazza F, Martinelli M, Colzani D, Beretta R, Zappa A, Orlando G. High performance of a new PCR-based urine assay for HPV-DNA detection and genotyping. J Med Virol. 2013 Jan;85(1):91-8. doi: 10.1002/jmv.23434.
à Bianchi S, Frati ER, Panatto D, Martinelli M, Amicizia D, Zotti CM, Martinese M, Bonanni P, Boccalini S, Coppola RC, Masia G, Meloni A, Castiglia P, Piana A, Gasparini R, Tanzi E. Detection and genotyping of human papillomavirus in urine samples from unvaccinated male and female adolescents in Italy. PLoS One. 2013 Nov 8;8(11):e79719. doi: 10.1371/journal.pone.0079719.
à Frati ER, Fasoli E, Martinelli M, Colzani D, Bianchi S, Carnelli L, Amendola A, Olivani P, Tanzi E. Sexually Transmitted Infections: A Novel Screening Strategy for Improving Women's Health in Vulnerable Populations. Int J Mol Sci. 2017 Jun 20;18(6):1311. doi: 10.3390/ijms18061311.
à Orlando G, Tanzi E, Chatenoud L, Gramegna M, Rizzardini G; VALHIDATE Study Group. Rationale and design of a multicenter prospective cohort study for the eVALuation and monitoring of HPV infections and relATEd cervical diseases in high-risk women (VALHIDATE study). BMC Cancer. 2012 May 30;12:204. doi: 10.1186/1471-2407-12-204.]

Reviewer 2 Report
I was invited to revise the paper entitled "The effect of HPV vaccination in Italy: type-specific prevalence evaluated in different cohorts a decade after implementation of the vaccination program". It aimed to evaluate the crude prevalence of HPV in three different cohort of Italian subjects.
The paper is very interesting and it can help policy makers to evaluate the impact and possible changes in HPV vaccination campaign.
Major observations:
- Sample size estimation is missing;
- Statistical analysis section is too poor. Statistical comparisons and relatives tests performed should be deeply described;
- P-values cannot be reported as <0.05 or N.S. Exact p-values should be reported;
- Authors should justify the reason of different number of enrolled patients across different cohorts. Differently, should better describe the enrollment methods performed;
- Enrolled patients are from different Italian Regions, so are vaccination programs different across these Regions? Authors in addition should stratify the analysis by Regions;
- A Table reporting baseline characteristics of enrolled patients should be presented.
Author Response
Reply to Reviewer 2
#REV2
We thank the Reviewer #2 for his comments. We agree with the Reviewer #2 and we have made almost all of the suggested corrections.
I was invited to revise the paper entitled "The effect of HPV vaccination in Italy: type-specific prevalence evaluated in different cohorts a decade after implementation of the vaccination program". It aimed to evaluate the crude prevalence of HPV in three different cohort of Italian subjects.
The paper is very interesting and it can help policy makers to evaluate the impact and possible changes in HPV vaccination campaign.
Major observations:
- Sample size estimation is missing.
The sample size estimation was added to the manuscript (Material and Methods section) and was calculated on the basis of published data on the prevalence of HPV infection in the Italian population aged 18-40 years.
- Statistical analysis section is too poor. Statistical comparisons and relatives tests performed should be deeply described.
An in-depth description has been added as required.
- P-values cannot be reported as <0.05 or N.S. Exact p-values should be reported.
As suggested, exact p-values have been reported.
- Authors should justify the reason of different number of enrolled patients across different cohorts. Differently, should better describe the enrollment methods performed.
In the Conclusions section a sentence on the discrepancy between the number of enrolled patients across different cohorts.
- Enrolled patients are from different Italian Regions, so are vaccination programs different across these Regions? Authors in addition should stratify the analysis by Regions.
As the HPV vaccination strategy was the same across the Country without difference among Regions, a comprehensive analysis was conducted.
- A Table reporting baseline characteristics of enrolled patients should be presented. As suggested, a table showing the demographic characteristic of the study population has been added.

Round 2
Reviewer 1 Report
As pointed out by this referee, authors have clearly recognized limitations regarding the sample size of their work, so to consider it as a pilot study. Some sentences stating the possible limitations due to using exclusively urine samples would be appreciated. Authors have sufficiently improved the mansucript to deserve publication.
Author Response
We thank the Reviewer 1 for his comments. We agree with Reviewer 1 and we added a sentence stating the possible limitations due to using exclusively urine samples
Reviewer 2 Report
Authors addressed all comments raised.
Author Response
We thank the Reviewer 2 for his approval.
The English version has been revised.